# Inflammation-Driven Colorectal Cancer Associated with Colitis: From Pathogenesis to Changing Therapy

**DOI:** 10.3390/cancers15082389

**Published:** 2023-04-20

**Authors:** Olga Maria Nardone, Irene Zammarchi, Giovanni Santacroce, Subrata Ghosh, Marietta Iacucci

**Affiliations:** 1Department of Public Health, University Federico II of Naples, 80131 Naples, Italy; olgamaria.nardone@unina.it; 2Department of Medicine, University College of Cork, T12 R229 Cork, Ireland; izammarchi@ucc.ie (I.Z.); gsantacroce@ucc.ie (G.S.); subrataghosh@ucc.ie (S.G.)

**Keywords:** carcinogenesis, colitis-associated cancer, inflammation, IBD therapy, gut barrier, OMIC, microbiome

## Abstract

**Simple Summary:**

Patients with inflammatory bowel disease (IBD) are at an increased risk of developing colorectal cancer (CRC), mainly because of chronic intestinal inflammation. Some unique molecular differences occur in colitis-associated CRC, resulting in a different sequence of events, primarily of inflammation–dysplasia–carcinoma, compared to sporadic cases. In this context, the recent and continuous evolution in our understanding of the genetic and molecular mechanisms underlying IBD-associated CRC and the development of new OMIC techniques has opened up new possibilities for personalised and organ-sparing therapies. This review provides an overview of the IBD-related carcinogenesis pathway with a focus on the interaction between microbiota and the gut barrier and the role of currently available therapies for IBD in the inflammation–dysplasia–cancer sequence.

**Abstract:**

Patients affected by inflammatory bowel disease (IBD) have a two-fold higher risk of developing colorectal cancer (CRC) than the general population. IBD-related CRC follows a different genetic and molecular pathogenic pathway than sporadic CRC and can be considered a complication of chronic intestinal inflammation. Since inflammation is recognised as an independent risk factor for neoplastic progression, clinicians strive to modulate and control disease, often using potent therapy agents to achieve mucosal healing and decrease the risk of colorectal cancer in IBD patients. Improved therapeutic control of inflammation, combined with endoscopic advances and early detection of pre-cancerous lesions through surveillance programs, explains the lower incidence rate of IBD-related CRC. In addition, current research is increasingly focused on translating emerging and advanced knowledge in microbiome and metagenomics into personalised, early, and non-invasive CRC screening tools that guide organ-sparing therapy in IBD patients. This review aims to summarise the existing literature on IBD-associated CRC, focusing on new insights into the alteration of the intestinal barrier and the interactions with the gut microbiome as the initial promoter. In addition, the role of OMIC techniques for precision medicine and the impact of the available IBD therapeutic armamentarium on the evolution to CRC will be discussed.

## 1. Introduction

Inflammatory bowel disease (IBD), comprising ulcerative colitis (UC) and Crohn’s disease (CD), are chronic relapsing inflammatory disorders in which a complex interplay of genetic predisposition and environmental factors alter host–microbiota interactions and lead to dysregulation of gut immune responses in a genetically susceptible individual [1]. The literature has widely consolidated that IBD is associated with an increased risk of colorectal cancer (CRC) compared to the general population [2,3]. A meta-analysis of 116 studies by Eaden et al. [3] reported that the incidence of IBD-associated CRC was 2%, 8%, and 18% at 10, 20, and 30 years after the onset of UC, respectively. Concerning CD, the risk of CRC is debatable and considered slightly elevated compared to the general population. A subsequent meta-analysis [4] estimated a cumulative risk of 2.9% at 10 years, 5.6% at 20 years, and 8.3% at 30 years in patients with CD. Several risk factors are associated with the development of colonic neoplasia, including longer disease duration, greater extent of colonic involvement, a family history of CRC, primary sclerosing cholangitis, male sex, and younger age at diagnosis. There is also additional risk related to worse disease severity, including high inflammatory burden, backwash ileitis, pseudopolyps, prior dysplasia, and colonic strictures [2]. However, recent evidence suggests that the incidence rates of IBD-associated CRC decreased over the last decade due to better control of inflammation and advances in the endoscopic detection and resection of precancerous lesions which have improved surveillance programs [5]. Nevertheless, it should be noted that most of these data came from tertiary referral centres wherein the study population was selected. 

IBD-associated CRC (IBD–CRC) has been considered a distinct entity compared to sporadic CRC (sCRC), and little is known about the pathogenesis and mechanisms behind the IBD–CRC [6]. The key driver of neoplastic changes and progression is chronic inflammation that contributes to dysplasia and is considered the most critical risk factor for developing colitis-associated CRC [7]. 

Chronic inflammation generates oxidative stress-induced DNA damage that may activate tumour-promoting genes and inactivate tumour-suppressing genes [8]. As a result, markers of oxidative damage and DNA double-strand breaks increase progressively in the inflammation–dysplasia–carcinoma sequence rather than the ‘adenoma–carcinoma’ sequence typical of sCRC (Figure 1). In addition, recent studies have demonstrated a potential role for the gut microbiome and host immune system. These lead to subsequent events that cause genetic and epigenetic alterations followed by clonal expansion of somatic epithelial cells, influenced by surrounding stromal and immune cells [9]. 

Here, we aim to provide an overview of the molecular pathway of IBD-related CRC with a particular focus on the role of multi-omics in the carcinogenesis process and the impact of medical therapy on developing and preventing CRC.

## 2. IBD–CRC: A Distinct Molecular Pathway 

The molecular pathways occurring in IBD–CRC differ from those observed in sCRC. The adenoma-to-carcinoma pathway prevails in sporadic cancer, while the sequence inflammation–dysplasia–carcinoma characterises the colitis related-CRC (Figure 1).

More in-depth, constitutive activation of WNT/b-catenin signalling via loss-of-function mutations in APC is the primary and earliest player of sCRC development, regulating cell fate, proliferation, polarity, and stemness via b-catenin-dependent and independent mechanisms [10]. Multiple other driver genes, such as KRAS, P53, PIK3CA, SMAD4, ARID1A, MYC, etc., are involved in the following progression of sCRC. These genes are also involved in IBD–CRC even though the timing and frequency of some of the common gene alterations are different [11,12]. Notably, mutation and loss of function occur frequently and very early in the process, even before dysplasia, suggesting an alternative mechanism of WNT pathway activation [6]. Moreover, KRAS mutations occur less frequently and later in IBD–CRCs; nuclear accumulation of b-catenin is prevalent compared to sCRCs.

Interestingly, a “Big Bang” model of CRC evolution has been formulated to support this concept. Chronic inflammation induces tumour-promoting molecular alterations in pre-existing clonal cell populations that occur sharply rather than a gradual accumulation driven by pressures from the microenvironment [6,13]. 

A powerful tool to describe tumour transcriptional, genetic, epigenetic, and microenvironment characteristics is a transcriptome-based classification of CRCs that consists of four consensus molecular subtypes (CMSs).

CMS1 (microsatellite instability immune, overall 14%) characterised by hypermutated, microsatellite unstable and strong immune activation; CMS2 (canonical, overall 37%) marked by epithelial cell differentiation with upregulated of WNT and MYC signalling activation; CMS3 (metabolic, overall 13%), with epithelial and evident metabolic dysregulation; and CMS4 (mesenchymal, overall 23%) mediate the activation of the pathway related to the epithelial–mesenchymal transition and stemness with transforming growth factor–β activation and a prominent stromal invasion [11,14]. 

Regarding IBD–CRC, the CMS distribution has yet to be fully clarified even though a complete lack of CMS2 tumours and a skewing towards the CMS4-associated epithelial–mesenchymal transition pathway is the most prevalent. These are associated with EMT, matrix remodelling, transforming growth factor b (TGF-b) signalling, complement activation, and depletion of WNT/MYC-related expression signatures [6]. 

Rajamäki K et al. [11] have recently analysed whole-genome sequencing, single nucleotide polymorphism arrays, RNA sequencing, genome-wide methylation analysis, and immunohistochemistry from 31 patients with IBD–CRC. 

Notably, they reported a complete absence of canonical epithelial tumour subtypes associated with WNT signalling CMS2 tumours and a predominance of the mesenchymal tumours related to oncostatin M receptor overexpression CMS4 among IBD–CRCs

Additionally, the propensity for a relative loss in expression of HNF4, a transcription factor essential for the embryonic development of colonic epithelium and the overexpression of a receptor for cytokine oncostatin M (OSMR), which may contribute to a more mesenchymal phenotype was observed. 

Elevated intestinal OSMR and OSM expression are also associated with a subgroup of IBD patients showing poor response to TNFa blockers. Hence, OSM could be considered a potential biomarker and therapeutic target for IBD. However, further studies are required to establish the non-invasive detection of colorectal adenomas and carcinomas from biomarkers.

## 3. OMICS: Future Directions

Multi-omics studies consisting of meta transcriptomics, metaproteomic, and metabolomics offer a great chance to fill knowledge gaps in IBD pathogenesis for a better understanding of CRC development as well as its classification into different molecular subtypes for patient stratification and the development of new biomarkers and targeted therapies [15] (Figure 2).

Further OMICs approaches allow better characterisation of the gut microbiome and how the microbiome in turn influences the surrounding ecosystem in order to identify why some IBD patients develop CRC and others do not [16]. 

However, most studies are based on individuals with sporadic CRC, whilst there is still a relatively small number of studies integrating omic datasets from IBD patients. The integrative human microbiome project (iHMP) collects host and microbiome-associated data using multiple meta-omics strategies [17,18]. Specifically, this project aims to identify microbial community changes over time that precedes human diseases such as IBD [19].

More recently, a comprehensive multi-omics project on the pathogenesis and outcomes of primary sclerosing cholangitis (PSC) associated with IBD was developed. Based on the hypothesis that multi-omics analyses of data capturing environmental exposures and the associated biological responses, the project aims to better understand the role and interplay of the genome, exposome, and microbiome on pathways influencing PSC pathogenesis and outcomes [20]. 

## 4. Primary Sclerosing Cholangitis (PSC) as a Risk Factor for IBD–CRC

Primary sclerosing cholangitis (PSC) is a chronic cholestatic liver disease with a pooled prevalence in patients with IBD of 2.16%. When low-grade dysplasia is detected in patients with PSC–IBD, they are at higher risk of developing advanced CRC compared to patients with IBD without PSC, suggesting a more aggressive disease course [2].

Hence, in IBD with concomitant primary sclerosing cholangitis (PSC) the risk of CRC increases substantially, yielding a 3-to 4-fold higher risk compared with having IBD alone [21]. In an attempt to explain this augmented risk, in a recent study, de Krijger M et al. [22] analysed DNA copy number variations, microsatellite instability (MSI), mutations on 48 cancer genes, and the CpG island methylator phenotype (CIMP) from resection specimens of 19 patients with PSC–IBD–CRC. Furthermore, they compared these genetic profiles with two published cohorts of IBD-associated CRC (IBD–CRC; *n* = 11) and sporadic CRC (s-CRC; *n* = 100). The excess risk of CRC in patients with PSC–IBD could not be explained by copy number aberrations, mutations, MSI, or CIMP status in cancer tissue or in adjacent mucosa.

Most importantly, no significant differences were found in patterns of chromosomal aberrations in PSC–IBD–CRC concerning those observed in IBD–CRC and s-CRC. Mutation frequencies were similar between the groups, except for mutations in KRAS, which were less frequent in PSC–IBD–CRC (5%) versus IBD–CRC (38%) and sCRC (31%; *p* = 0.034) and in APC which were less frequent in PSC–IBD–CRC (5%) and IBD–CRC (0%) versus sCRC (50%; *p* < 0.001). Notable cases of PSC–IBD–CRC were frequently CIMP positive (44%), similar to sCRC (34%; *p* = 0.574), whilst they were less frequent than in patients with IBD–CRC (90%; *p* = 0.037) suggesting that epigenetic changes may play an important role. Hence, these findings pave the way for further exome-wide and epigenetic studies.

## 5. The Role of the Intestinal Barrier in Colorectal Cancer Development 

The interplay between the intestinal barrier and microbiome is crucial in IBD pathogenesis. The impairment of the intestinal barrier (especially of the epithelium) and a concomitant microbiome alteration (defined as ‘dysbiosis’) are considered the cornerstones of disease development [23,24]. While it is unclear which is the primum movens in this process, what is certain is that the alteration of mucosal barrier integrity, with the impairment of tight junctions (TJ) and the interaction with a harmful microbiome act as triggers for immune processes in the lamina propria, activating inflammatory immune processes, with possible cancer development (Figure 3).

Under physiological conditions, if epithelial barrier function is preserved, commensal bacteria interact with pattern recognition receptors (PRRs) in the apical membrane of enterocytes, activating a process of microbial tolerance through the inhibition of NF-kB signalling and the induction of a tolerogenic phenotype of dendritic cells (DCs) and macrophages [25,26]. Conversely, in the case of dysbiosis and intestinal barrier impairment, the activation of PRRs on basolateral membrane determines the activation of the inflammatory pathway [27,28]. More in-depth, the interaction of microbial antigens with lamina propria DCs and macrophages directs the inflammation through the production of inflammatory cytokines (TNF-α, IL-12 and IL-23) and the consequent activation of innate immune cells, such as neutrophils and eosinophils. In addition, the interaction between DCs and microbes activates the adaptative (Th1 and Th17) immune response in mesenteric lymph nodes. Through the vascular system, activated T cells reach the lamina propria, spreading inflammation by producing additional inflammatory cytokines (such as IFN-gamma, IL-17, and IL-22). As previously discussed, the persistence of inflammation can lead to carcinogenesis and metastasis.

### 5.1. Microbiome Interaction in Colorectal Cancer Development

In recent years, the role of bacteria in the carcinogenic process has been extensively explored in several studies through in vitro cell culture, intestinal organoid models, and mouse models of inflammation. 

Immuno-deficient mouse models of CRC supported the mechanisms of microbiota-induced inflammatory tumorigenesis and susceptibility between individuals depending on Toll-Like Receptor (TLR)/MyD88 and inflammasome signalling [29]. 

Currently, three bacterial species have been linked in particular to the process of human colorectal carcinogenesis: *Fusobacterium nucleatum* (Fn), *Escherichia coli* containing pathogen polyketide synthetase (pks) islands, and *Bacteroides fragilis* expressing *B. fragilis* toxin (BFT) [6].

The intestinal microbiota of patients with IBD demonstrates a greater abundance of Enterobacteriaceae/*E. coli* and patients with IBD and CRC have an increased prevalence of mucosa-associated *E. coli* compared with non-IBD and non-CRC patients.

*E. coli* strains that contain the pks gene cluster have been found more often in biopsies from CRC (67%) and IBD (40%) than in healthy control subjects (21%). These results in oncogenic phenotypes manifest in WNT independence and increased proliferation. The pks encodes colibactin, a genotoxin that can stimulate tumour growth. In mucosal inflammation, pks+ *E. coli* promoted DNA damage and neoplastic transformation in a mouse model [30]. This suggests a direct link between colibactin exposure and increased cancer risk.

Moreover, Enterotoxigenic *Bacteroides fragilis* expresses the pathogenic BFT which binds to a specific colonic epithelial cell receptor, thus activating Wnt and NF-kB signalling pathways that determined increased cell proliferation, the epithelial release of pro-inflammatory mediators, and DNA damage. In 90% of patients with sporadic CRC, the BFT gene sequences were observed compared with 55% of controls and in the stool of approximately 14% of patients with IBD. Of note, BFT induces acute and chronic colitis in mice and it promotes IL-17-dependent colon carcinogenesis in the Min Apc+/− mouse model [31,32]. 

Furthermore, metabolites such as short-chain fatty acids produced by the commensal GI microbiota modulate immune cell activation, inflammatory responses, and carcinogenesis via tumour suppressor gene expression and regulatory T-cell proliferation through histone deacetylase inhibition [16]. Other metabolic classes and pathways significantly dysregulated in CRC include bile acids [33]. Experimental mouse models have shown that elevated secondary bile acid concentrations promote intestinal tumorigenesis [16]. Wirbel et al. [34] found the bile acid-inducible (bai) operon to be highly abundant in the stool of CRC patients. They confirmed this finding at both the genomic and transcriptomic levels using qPCR. Moreover, they supported a link between high dietary fat intake and CRC [35]. Together, all these data strongly support the promising role of microbiome-based CRC diagnostics. In the future, identifying disease-specific microbiome signatures for both IBD and CRC, together with metagenomics sequencing and other culture-independent technologies, may be helpful for personalised, early, and noninvasive CRC screening in IBD patients [36].

### 5.2. Markers of Gut Barrier Functionality for the Early Detection of CRC

Considering the main role of intestinal barrier disruption in CRC development, markers evaluating its function have been considered for early cancer detection. 

Proteins involved in maintaining gut barrier function, such as intestinal fatty acid binding protein iFABP, have been proposed as potential biomarkers for detecting early-stage colorectal cancer (CRC) or assessing the malignant potential of adenomas [37]. iFABP is found in intestinal epithelial cells and its leakage into circulation during intestinal damage can upregulate its expression. Interestingly, plasma levels of iFABP are higher in patients with severe UC than in those with mild disease [38].

In addition to iFABP, tight junction proteins such as claudins (CLDNs) and junctional adhesion proteins (JAMs) have also been implicated in the pathogenesis of gastrointestinal cancers [39]. The overexpression of CLDNs has been linked to neoplastic transformation in premalignant epithelial tissue, while lower expression of junctional adhesion proteins (JAM2) is associated with CRC progression, metastasis, and poor prognosis [40]. Examining and identifying colonic mucosal barrier integrity markers may aid the early detection and prediction of the progression of CRC-associated IBD. Probe confocal laser endomicroscopy (pCLE) is an in vivo histology imaging tool capable of assessing ultrastructural and dynamic changes in the intestinal barrier consisting of epithelial cells connected by tight and adherent junctions.

This technique could predict the therapeutic response, thus paving the way to precision medicine [41]. The combination of innovative endoscopic techniques able to study intestinal anatomy and cell function as well as new OMICs techniques, termed “Endo-Omics, will in the future enable a better understanding of IBD, the optimisation of therapeutic management, and cancer prevention [42].

## 6. Selective and Targeted IBD Therapies and Their Anticarcinogenic Role

A commonly acknowledged goal in managing inflammatory bowel disease (IBD) is to effectively control inflammation and thus induce and maintain deep remission [43,44]. Several studies have emphasised the correlation between decreased inflammatory activity and a reduced risk of developing CRC [45,46]. Over the last decades, biological therapies have revolutionised the treatment of IBD. In this context, the role of therapies has been controversial as it requires a careful assessment of the ratio between the potential benefits and risks.

It has been hypothesised that the chemopreventive properties of the main classes of drugs available for IBD treatment are due to their direct (anticarcinogenic mechanisms) and indirect activity (reducing inflammatory activity) [47,48]. The scientific community has recently questioned the risk of developing colorectal cancer in IBD patients after being treated with immunomodulator therapy. However, precise and individual long-term safety profiles of biologicals, such as monotherapy or in combination with immunomodulators, are poorly studied. In this context, therapies are considered a double-edged sword in IBD–CRC. While the curative role of immunosuppressive therapies on chronic intestinal inflammation is supposed to reduce the risk of colitis-associated CRC, a secondary neoplasm is one of the most feared sequelae of immune system manipulation [49]. 

Several studies using different drug classes have yielded conflicting results, ranging from a protective effect to a negative or absence of an effect. These negative findings are increasingly attributed to methodological differences. Randomised controlled trials are often lacking in this field and most studies are observational, including retrospective and prospective cohort or case-control studies. To overcome these problems, new drugs are becoming more and more selective, thus reducing the risk of general immunosuppression. 

### 6.1. 5-ASA Compounds

Aminosalicylates (5-ASA), including mesalazine, sulfasalazine, olsalazine, and balsalazide are among the oldest therapies currently used to manage IBD. However, despite new treatment options, 5-ASA compounds remain the effective first-line therapy for a step-up approach in treating mild to moderate UC and for maintenance of remission [44]. Remarkable, lifelong treatment with 5-ASA is recommended as it has a chemopreventive effect attributed to both direct anticarcinogenic and anti-inflammatory effects. The immediate anticarcinogenic effect is due to the medication’s interference with multiple pathways related to DNA replication, response to damage, cell growth and proliferation, carcinogenesis, and tumour signalling.

Of note, there are several mechanisms involved, including the modulation of proteins that control the cell cycle [50], inhibition of molecules that regulate angiogenesis [51,52], scavenging of molecules that increase DNA oxidative stress, and reduction of β-catenin accumulation in APC-mutated cells [53,54]. In addition, 5-ASA also suppresses the Wnt/β-catenin pathway, which is an effective anticarcinogenic mechanism. Finally, it has also been found to induce a cell cycle stop or apoptosis in carcinomatous cells with deficient/mutated β-catenin or mutated COX-2 [47].

The relationship between 5-ASA use and the risk of CRC has been widely investigated in clinical studies. In a recent meta-analysis, a patient who had been treated with 5-aminosalicylic acid (5-ASA) for their reduced risk of advanced colorectal neoplasia (aCRN), as indicated by a pooled univariable odds ratio (OR) of 0.53 (95% CI, 0.39–0.72; I^2^ = 67%). Furthermore, six studies that provided multivariable ORs also showed a lower risk (pooled OR, 0.51; 95% CI, 0.39–0.66). However, most included studies had a retrospective design [55].

Several studies found no or minimal protective effect of sulfasalazine for dose >2 g/die [55,56,57]. Accordingly, the central hypothesis for this was lack of efficacy and poor adherence to treatment. Improving patients’ knowledge is crucial as it could enhance adherence given that current adherence rates have been reported to be around 40% [58].

### 6.2. Thiopurines

Both 6-mercaptopurine and azathioprine are approved as maintenance treatment for steroid-dependent or refractory IBD [59,60]. Before the widespread use of biological agents, they were administered in about 50% and 20% of CD and UC patients, respectively [61]. Currently, they are mainly used in addition to monoclonal antibodies, such as anti-tumour necrosis factor-α and anti-α4β7 inhibitors, to prevent immunogenicity [43,44] and to increase their efficacy. Studies evaluating the chemoprotective effect of thiopurines have had discrepant results, with most showing no benefit in chemoprevention. 

The CESAME (Cancers Et Surrisque Associé aux Maladies inflammatoires intestinales en France) was the first nationwide prospective observational cohort that included 19,486 patients designed to assess the possible excess risk of cancer in patients with IBD receiving thiopurines. Noteworthy patients receiving thiopurines have an increased risk of developing lymphoproliferative disorders [62]. Subsequently, a case-control study nested in the CESAME cohort showed that a significant decrease in the risk of colorectal cancer in IBD was associated with exposure to aminosalicylates but not to thiopurines [63]. A recent large meta-analysis with a total of 27 studies showed that thiopurine use reduced the risk of CRC, particularly in patients with an IBD history of at least 8 years; however, subgroup analysis based on the geographical location revealed that no protective effect was observed in studies conducted in North America [64]. These results align with a retrospective cross-sectional study conducted by Alkhayyat et al. which found a statistically insignificant correlation between thiopurine usage and CRC in patients with CD and a slight elevation in the risk of CRC in patients with UC [65]. Several studies showed controversial results; they may be chemopreventive for CRN but not for ACN/CCR [66] while others found that they were protective for ACN [67,68]. A systematic review and meta-analysis revealed that thiopurines may have a protective effect against the development and progression of colorectal neoplasia (dysplasia and/or cancer), particularly in cases where treatment exceeds six months, disease duration surpasses eight years, and hospital-based studies are conducted. Additionally, the protective effect against LGD progression was more prominent in studies that excluded patients with primary sclerosing cholangitis. However, there seems to be some discrepancy in the research: another study found that the antineoplastic effect primarily targeted ACN/CCR rather than low-grade dysplasia and another review did not discover a significant protective effect on the risk of colorectal neoplasia (dysplasia and/or cancer). Finally, several studies investigated the side effects of long-term thiopurine therapy, revealing an increased risk of extraintestinal cancer (lymphoma and non-melanoma skin cancer) [69]. Consequently, the long-term use of thiopurines should not be recommended for colorectal cancer prevention.

### 6.3. Anti-TNF α Agents

Although anti-TNFα agents have been introduced in clinical practice for many years, studies evaluating their chemoprotective effect are scarce. Data are mainly extrapolated from works about long-term safety concerns. Their anticarcinogenic effect seems mainly due to their anti-inflammatory properties by inducing mucosal healing. However, a direct antineoplastic role can be hypothesised.

TNF-α is a cytokine that promotes apoptosis of intestinal epithelial cells and activates the NF-kB transcription pathway, thereby influencing the activity of innate immune cells [70]. TNFα can promote early epithelial alterations observed in intestinal metaplasia-dysplasia, inducing epigenetic changes and increasing oncogene expression levels [71]. In mouse models, TNFα levels were raised in the mucosa and submucosa before CAC was developed. Treatment with anti-TNFα antibodies significantly improved mucosal healing, with fewer tumour lesions than controls [72]. 

The effect of anti-TNF treatment in real-life observational studies has yet to be extensively examined, particularly in population-based settings. However, two cohort studies that followed IBD patients treated with infliximab for a long-term follow-up did not report an increased incidence of colorectal cancer [73,74]. Furthermore, data from a multicenter US large-scale database study revealed a significant reduction in the risk of colorectal cancer among patients receiving anti-TNF agents in both CD (OR 0.69; 95% CI, 0.66–0.73) and UC (OR 0.78; 95% CI, 0.73–0.83), as well as in combined treatment (anti-TNFα and immunomodulators) in CD (OR 0.73; 95% CI, 0.63–0.84) and UC (OR 0.83; 95% CI, 0.70–0.99) [65]. 

In a recent study from France, the risk of colorectal cancer was only reduced in patients with long-standing colitis (more than 10 years) (HR 0.41; 95% CI, 0.20–0.86) and not in all patients treated with anti-TNF [75]. Another study that considered patients who underwent liver transplantation for primary sclerosing cholangitis and were treated with infliximab or adalimumab revealed that 3 patients developed colorectal cancer, despite 64% of patients showing endoscopic improvement and 21% achieving remission. However, due to concomitant immunosuppressive treatment and short-term therapy with anti-TNFα, clear conclusions could not be drawn [76]. Thus, decisions on the use and timing of treatment with TNFα antagonists could be made case by case. 

### 6.4. Anti-Lymphocyte Trafficking Agents 

Vedolizumab is a humanised IgG1 monoclonal antibody that selectively blocks α4-β7 integrin, preventing the translocation of T cells from vessels to the intestinal and colonic mucosa. This mechanism of action leads to a drastic drop in mucosal inflammation, making it effective both for induction and maintenance of the clinical response and remission [77,78]. Given the local immunosuppressive effect, there was a concern that vedolizumab may increase the risk of epithelial cell dysplasia. However, some studies have suggested that it may reduce the risk of colorectal cancer by promoting mucosal healing through their local anti-inflammatory activity rather than through a direct anti-carcinogenic effect.

The initial study examining the impact of vedolizumab treatment on colonic dysplasia incidence was conducted retrospectively on patients enrolled in the extension phase of the GEMINI trials. These patients received 300 mg of vedolizumab every 4 weeks for over a year. The long-term safety study also included patients who were not previously randomised in the GEMINI trials but received open-label induction therapy with vedolizumab.

Endoscopic/combined healing was reached in 29%/21% CD and 50%/32% of UC patients, respectively. Surveillance colonoscopies highlighted that low-grade dysplasia in targeted biopsies (obtained from areas with abnormal mucosal appearance) was observed in four UC and two CD patients (10%) and none progressed to high-grade dysplasia or cancer. The only case of high-grade dysplasia was identified in the colectomy specimen of a UC patient with low-grade dysplasia at biopsies and recurrent moderately active endoscopic and histological colitis [79]. 

Some studies report no increased risk for CRC [80,81] in patients with a history of cancer [82]. In a retrospective observational cohort study, after a median follow-up period of 19.4 [14.0–29.9] months, 9 out of 75 patients with IBD and PSC treated with vedolizumab developed CRC. However, it is essential to bear in mind that PSC itself is a risk factor for CRC [83]. Finally, the Vedolizumab Global Safety Database has reported post-marketing surveillance data for patients with CD or UC treated with vedolizumab, notably with over four years of follow-up. These medium-term safety data, covering 208,050 patient years of vedolizumab exposure, showed no overall increased signals for malignancy in either CD or UC patients [84].

Other therapies targeting leukocyte trafficking, such as etrolizumab and ontamalimab, are under investigation. However, although preliminary data showed no increased cancer risk, more data are needed [85,86].

### 6.5. Targeting the IL12/IL23 Axis 

IL23 is a cytokine produced by immune cells and is involved in the pathogenesis of IBD and CRC [87]. Its production is increased in the mucosa of mice with colitis and CAC, and it is associated with an increased level of transcription of BATF, which can induce Th17 cells. It is involved in the development of colitis-associated colon tumours [88].

The suppression of the function of its receptors (IL12/IL23 p40, IL23 p19, or IL23R) leads to a marked decrease in colonic inflammation by reducing the activation of immune cells usually stimulated by this cytokine (mainly, Th 17 cells, granulocytes, and NK cells) together with the reduced production of proinflammatory cytokines. 

Studies on mouse models showed that knocking out molecules involved in IL23 production led to protection from colitis-associated tumours [89] and decreased levels of antitumorigenic IL12 [90]. Although other studies are needed, targeting IL23 emerges as an important concept for suppressing gut inflammation and inflammation-associated cancer growth. Consistently, neutralising antibodies against IL12/IL23 p40 and IL23 p19 have been successfully used in clinical trials for the therapy of CD [91] and UC [92].

Current evidence does not show an increased overall cancer risk in IBD patients treated with anti-IL12/23 agents. However, long-term data are lacking in IBD. The IM-UNITI program followed up with patients with CD for up to 5 years and found no increased risk of malignancy associated with ustekinumab treatment. In the initial 44 weeks, there were 8 cases of non-melanoma skin cancer (NMSC) across the entire study population with no significant differences between patients receiving ustekinumab or placebo. From weeks 44 to 96, the rate of treatment-emergent malignancies per 100 patient-years was 2.60 for the placebo and 0.37 for ustekinumab, though the results are limited by the lack of an adequate reference group as only 61 patients continued with the placebo beyond week 44 in the long-term extension study [93]. With regard to UC, data from the UNIFI program with a follow-up period of 3 years revealed an incidence of malignancy per 100 years of follow-up of 0.72 [95% CI: 0.33–1.36] for ustekinumab and 0.66 [95% CI: 0.08–2.38] for the placebo. However, the limited size of the placebo reference group during the long-term extension study (which included only 115 stable patients receiving placebo) makes interpretation challenging [94]. Observational studies support these findings, suggesting that malignancies are rare. In a multicentre cohort of 142 CD patients, dose escalation of ustekinumab up to every 4 weeks did not increase the risk of adverse events, including malignancies. Over a follow-up period of up to 52 weeks, only one case of cervical intraepithelial neoplasia and NMSC were reported [95]. 

Other promising drugs targeting the IL-12/IL-23 axis are mirikizumab, guselkumab, and risankizumab. These selectively inhibit the p19 subunit of IL23 which seems to be related to a reduced malignant transformation of colonic lesions [96]. Despite limited real-life data, cancer risk does not seem to be increased [97]. Nevertheless, more prospective data are necessary to better understand this context.

### 6.6. Small Molecules

#### 6.6.1. Targeting the JAK/STAT Pathway

Tofacitinib is a small oral molecule able to mainly inhibit JAK1 and JAK 3. It was approved for treating ulcerative colitis in 2018 [48]. The activation of the JAK/STAT pathway determines the production of several proinflammatory cytokines. 

Thus, its inhibition can improve colitis and reduce the risk of colorectal cancer [49]. Real-lifee studies about its ability to reduce the risk of colorectal cancer are lacking due to the short follow-up time. However, an indirect anticarcinogenic effect can be hypothesised due to its anti-inflammatory action. However, so far there is no evidence that the overall risk of cancer is increased in patients with IBD treated with JAK inhibitors [98]. A meta-analysis of RCTs did not find any significant differences in the risk of NMSC between JAK inhibitors and either placebo or active comparator [RR: 1.21; 95% CI: 0.19–7.65] [99]. Real-world safety data on tofacitinib in UC patients are limited. In a study by Deepak et al. involving 260 UC patients, only 2 cases of malignancy were observed [100]. Another real-world cohort of 113 IBD patients showed that only one patient who had prior immunosuppressive exposure, developed a neoplasm [101]. Moreover, in the long-term extension phase of the OCTAVE study, only four patients receiving 10 mg of tofacitinib developed CRC during the follow-up period compared to zero cases among the controls [102]. 

Further prospective studies are needed to fully assess the risk of malignancy associated with JAK inhibitors in IBD. Currently, there are no data on the potential cancer risk of other Jak inhibitors being developed for IBD. Upadacitinib is a JAK inhibitor highly selective for JAK1. As well as the aforementioned drugs, data on reducing the risk of CRC can be hypothesized by its ability to induce endoscopic remission since real word data are upcoming [103]. In CD, the CELEST extension study showed a rate of endoscopic remission of 30% after 24 months of follow up and no cases of CRC [104]. In UC, the U-ACHIEVE maintenance study showed a rate of endoscopic remission of 25% at W52 and one case of CRC in the 30 mg group [105]. Furthermore, data from clinical trials of the filgotinib in rheumatoid arthritis show a similar incidence of malignancies compared to other JAK inhibitors [106,107]. The tyrosine kinase inhibitor decruvacitinib is still in phase II trials and its safety profile with respect to cancer risk is unknown [108]. 

#### 6.6.2. Sphingosine-1-Phosphate Receptor Modulators

S1P1 receptor agonists block the migration of lymphocytes from lymphoid organs by the internalisation and degradation of the S1P1 receptor [109]. Etrasimod and ozanimod are more selective S1P modulators. Ozanimod is an S1P1 and S1P5 receptor modulator licensed for UC, while etrasimod is a selective S1P1, S1P4, and S1P5 receptor modulator under investigation in UC and CD [108]. However, data concerning the risk of colorectal cancer require a long-term follow-up period; thus, they are not yet available [110] (Table 1).

## 7. Conclusions 

Patients with IBD have an increased risk of developing CRC, and the sequence of events leading to CRC in IBD differs from the sequence observed in sporadic cases. In IBD–CRC, inflammation plays a significant role in developing dysplasia and, ultimately, carcinoma. Consequently, preventing inflammation is crucial in minimising the risk of IBD–CRC. 

Estimating the risk of cancer associated with IBD therapy is complex as it is difficult to separate the risks associated with the therapy from those associated with the underlying disease. 

Therefore, an individualised assessment of the risk–benefit ratio of each therapy for each patient is crucial. 

While solid evidence supports the chemopreventive role of 5-ASA and anti-TNFα in IBD patients, there is still a lack of long-term data on the safety and efficacy of newer biological therapies such as vedolizumab, ustekinumab, and tofacitinib. Indeed, their long-term safety profiles, particularly with respect to cancer risk, are not yet fully understood.

The I-CARE (Inflammatory Bowel Disease Cluster for Assessment of Risk and Epidemiology) initiative is an important collaborative effort involving 15 countries and over 10,000 IBD patients to investigate the risks associated with IBD therapy, particularly biological therapies. It represents a significant step forward in understanding the risks and benefits associated with IBD therapy, particularly with respect to newer biological agents [112]. The data generated by this initiative will help inform clinical decision making and ultimately improve patient outcomes.

However, the decision to use or continue IBD therapy should be made on a case-by-case basis considering the patient’s individual risk factors, disease severity, and treatment goals.

The close monitoring and surveillance of patients receiving IBD therapy are still essential to detect any early signs of cancer development. Furthermore, the future of Endo-Omics will enable better understanding of the mechanisms of IBD cancer prevention and progression as well as early therapeutic management.

## Figures and Tables

**Figure 1 cancers-15-02389-f001:**
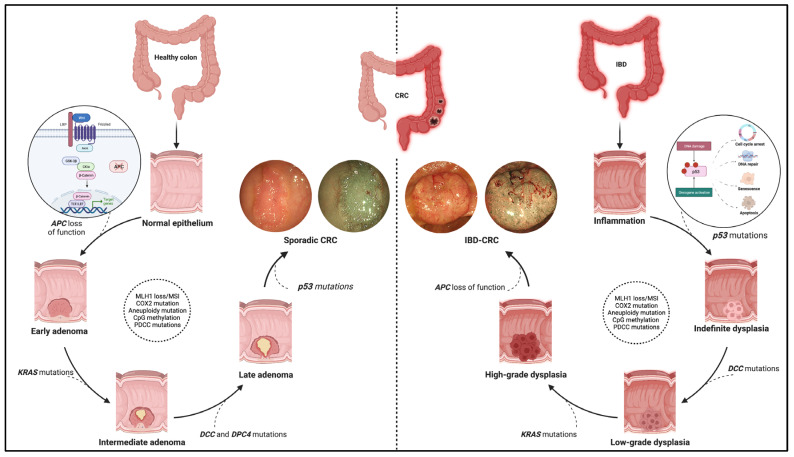
**Pathogenic pathway of sporadic colorectal cancer and IBD-related colorectal cancer.** The figure shows the different molecular pathways related to sporadic and IBD-related colorectal cancer (CRC). The sporadic CRC prevails in the adenoma-to-carcinoma sequence, while the inflammation–dysplasia–carcinoma cascade characterises the colitis related-CRC. In addition, the main gene mutations determining the progression of the two tumour phenotypes are reported with emphasis on the earliest mutations in the two processes. Namely, these entail APC loss of function, the WNT-beta catenin pathway activation for sporadic CRC (see zoom circle at left), and p53 mutations with consequent impacts on cell cycle, DNA repair, and cell viability for IBD-related CRC (see zoom circle at right). Finally, high-definition white light endoscopic images and virtual chromoendoscopic images (obtained through the Narrow Band Imaging technology) are provided for each phenotype of tumours. Created with “Biorender.com”. *Abbreviations: CRC, colorectal cancer; IBD, inflammatory bowel disease*.

**Figure 2 cancers-15-02389-f002:**
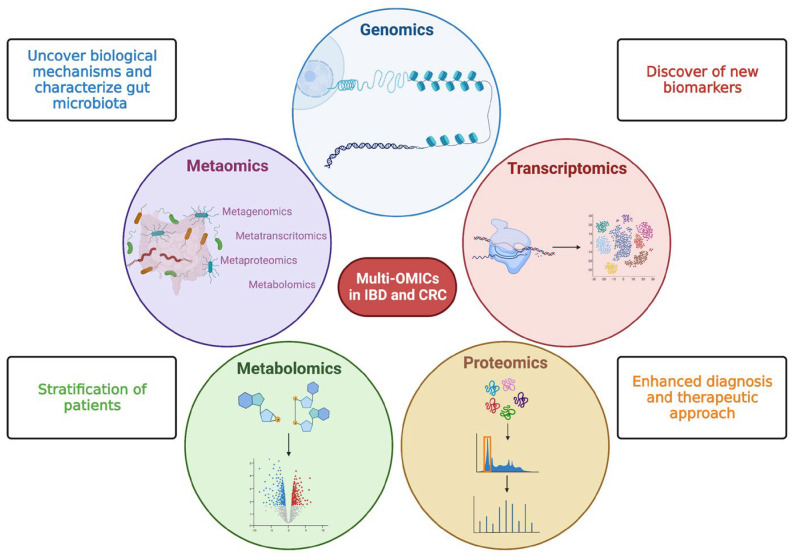
**Multi—OMICs and its impact on inflammatory bowel disease and colorectal cancer.** This figure schematically represents the main OMIC techniques available to date: genomics—the study of the genetic or epigenetic sequence information; transcriptomics—the evaluation of RNA transcripts; proteomics—the investigation of the structure and function of proteins; metabolomics—the identification and quantification of metabolites; metaomics—the application of the previously described techniques to the gut microbiome. The multiple and integrated application of these techniques, so-called multi-OMICs, will offer a great chance to fill knowledge gaps in inflammatory bowel disease (IBD) and colorectal cancer (CRC). In more detail, the application of multi-OMICs will provide (as specified in figure squares) the discovery of novel biological mechanisms below IBD and CRC pathogenesis, the detection of new clinically relevant biomarkers, the definition of integrated signatures able to stratify patients, and the enhancement of physician ability to establish disease prediction, establish a prognosis, and treat patients appropriately. Created with “Biorender.com”. *Abbreviations: CRC, colorectal cancer; IBD, inflammatory bowel disease*.

**Figure 3 cancers-15-02389-f003:**
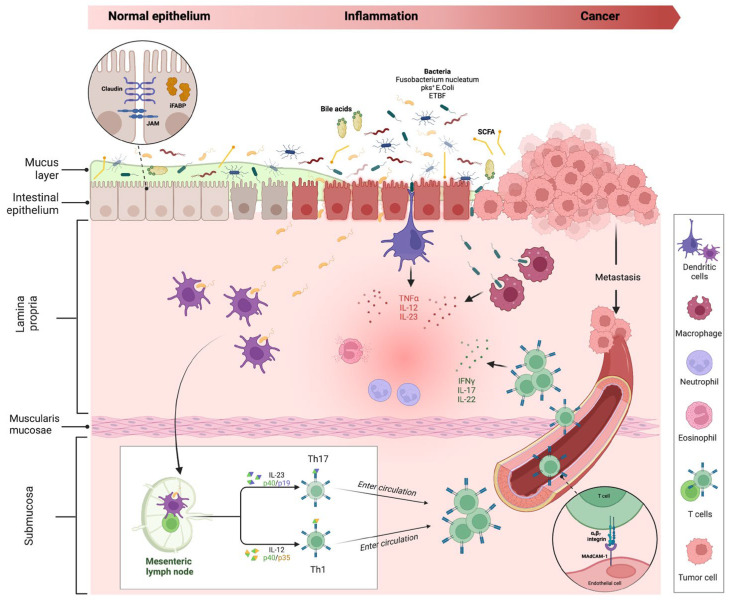
**Intestinal inflammation and evolution to cancer in IBD**. The impairment of the mucus layer and epithelial barrier, associated with dysbiosis, determines the inflammatory response, leading to IBD disease development and possible cancer evolution. Bile acids and small-chain fatty acids participate in initiating this process. In the lamina propria, dendritic cells and macrophages, after their interaction with intestinal microbes, determine the activation of innate immune cells through the release of numerous cytokines (neutrophils, eosinophils, etc.) and trigger the adaptative (Th1 and Th17) immune cells differentiation in mesenteric lymph nodes. Activated T cells, through a vascular homing mediated by the alfa4-beta7–MAdCAM-1 pathway, reach the intestinal lamina propria and spread the inflammatory process. The persistence of inflammation can lead to carcinogenesis and metastasis. Proteins involved in maintaining gut barrier function, such as intestinal fatty acid binding protein and tight junction proteins (shown in the dotted-line circle box in the upper left corner of the figure), have been proposed as potential biomarkers for cancer detection. Created with “Biorender.com”. *Abbreviations: E. coli, Escherichia coli; ETBF, Enterotoxigenic Bacteroides fragilis; iFABP, intestinal fatty acid binding protein; IFN, interferon; IL, interleukin; JAM, junctional adhesion molecule; MAdCAM, mucosal vascular addressin cell adhesion molecule; pks+, polyketide synthase productor; SCFA, small chain fatty acid; TNF, tumour necrosis factor*.

**Table 1 cancers-15-02389-t001:** Current and upcoming selective therapies for IBD.

Class	Molecule	Target	Mechanism of Action	Licensed	Data on CRC Risk
Anti-TNFα	Infliximab Adalimumab Golimumab	TNFα	Inhibition of the TNFα pathway	UC, CD	Not increased [73,74,75]
				UC	
Anti-lymphocyte trafficking agents	Vedolizumab	α4-β7 integrin	Prevention of translocation of T cells from vessels to the gut mucosa	UC, CD	Not increased * [80,81,82]
	Etrolizumab	β7 subunit of α4β7-αEβ7 integrins		No	Not available
	Ontamalimab	MAdCAM-1		No	Not increased ** [86]
IL12/IL23 axis	Ustekinumab	p40 subunit of IL23/IL12	Inhibition of the IL12-23 pathway	UC, CD	Not increased * [92,93,94]
	Risankizumab	p19 subunit of IL23	Inhibition of IL23 pathway	CD	Not increased ** [97]
	Mirikizumab	p19 subunit of IL23		No	Not available
	Guselkumab	p19 subunit of IL23		No	Not available
JAK inhibitors	Tofacitinib	JAK1 and 3	Reduced immune activation by inhibition of the JAK/STAT pathway	UC	Not increased * [102]
	Upadacitinib	JAK 1		UC, upcoming for CD	Not increased ** [103,105]
	Filgotinib	JAK 1		UC	Not increased ** [111]
	Deucravacitinib	Tyrosine kinase 2		No	Not available
S1P receptor modulators	Ozanimod	S1P1 and 5 receptor	Block the migration of lymphocytes from lymphoid organs	UC	Not increased ** [76]
	Etrasimod	S1P1, 4 and 5 receptor		No	Not available

Abbreviations: CD, Crohn’s disease; CRC, colorectal cancer; JAK, Janus kinase; IL, interleukin; MAdCAM, mucosal addressin cell adhesion molecule; S1P, sphingosine 1 phosphate; STAT, signal transducer and activator of transcription; UC, ulcerative colitis. * Data from long term safety studies and real-life studies, despite prolonged follow up, are required. ** Preliminary data from long term safety studies, more data are needed.

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
