# Peer review of "Inflammation-Driven Colorectal Cancer Associated with Colitis: From Pathogenesis to Changing Therapy"

_cancers, 2023, doi:10.3390/cancers15082389_

Round 1
Reviewer 1 Report
Overall this is an interesting and timely review. The landscape of treatments in IBD is changing and understanding the impact of these on the cancer development risks is important. This review serves to highlight areas of potential research and new opportunities as well as areas to approach with caution and where data is lacking.
The authors have covered the topic with sufficient depth to be of use to readers.
Main suggestion is that the authors restructure the manuscript to make better use of paragraphs. Many sentences are stand alone, while they should be formatted into paragraphs with the following sentences, I would like to suggest a minor re-write to amend this aspect of the style as this makes the manuscript disjointed and difficult to read.
Also, the authors mention endo-omics as a great opportunity to monitor patients but this is not discussed at much length within the manuscript. A section on the definition of this and potential would be useful.
Minor points:
1. Check use of CAC (section 4.5) is this distinct from CRC or interchangeable?
2. Use of IL12 and IL23 instead of IL-12, IL-23
3. Page 10, line 422 use of Baft - this should be BATF
4. Fig 2, spelling of epithelium at the top of the figure
5. Page 10 line 405; check this sentence, it looks incomplete.
6. Page 11 line 470; is this correct? The example of anti-cancer effect of the JAK inhibitor- more patients in the treatment arm developed cancer than the control?
Author Response
Overall this is an interesting and timely review. The landscape of treatments in IBD is changing and understanding the impact of these on the cancer development risks is important. This review serves to highlight areas of potential research and new opportunities as well as areas to approach with caution and where data is lacking.
The authors have covered the topic with sufficient depth to be of use to readers.
Main suggestion is that the authors restructure the manuscript to make better use of paragraphs. Many sentences are stand alone, while they should be formatted into paragraphs with the following sentences, I would like to suggest a minor re-write to amend this aspect of the style as this makes the manuscript disjointed and difficult to read.
Also, the authors mention endo-omics as a great opportunity to monitor patients but this is not discussed at much length within the manuscript. A section on the definition of this and potential would be useful.
Response: Thank you for your comment. We have now revised extensively the manuscript and formatted in several paragraphs. We also provided a clear definition of OMICs and included a section entitled “OMICs and Future directions”. Page 4-5
Minor points:
- Check use of CAC (section 4.5) is this distinct from CRC or interchangeable?
Response: Thank you for your comment, it means colitis associated colorectal cancer. We have now expanded the abbreviation.
- Use of IL12 and IL23 instead of IL-12, IL-23
Response: we have now edited it
- Page 10, line 422 use of Baft - this should be BATF
Response: We have now corrected it
- Fig 2, spelling of epitheliumat the top of the figure
Response: We have now edited it
- Page 10 line 405; check this sentence, it looks incomplete.
Response: We have now revised it: In a retrospective observational cohort study, after a median follow-up period of 19.4 [14.0-29.9] months, 9 out of 75 patients with IBD and primary sclerosing cholangitis treated with vedolizumab developed colorectal cancer. However, PSC itself is a risk factor for CRC. Page 13
- Page 11 line 470; is this correct? The example of anti-cancer effect of the JAK inhibitor- more patients in the treatment arm developed cancer than the control?
Response: we thank the reviewer for the insightful comment. We revised this paragraph and included the recent ECCO Guidelines on Inflammatory Bowel Disease and Malignancies, JCC 2022. Page 15
Reviewer 2 Report
The manuscript by Nardone et al describes what is known about the influence of IBD in CRC progression. It also summarizes data from different therapies that are being used in IBD.
I found the manuscript quite interesting and well illustrated but:
1. It is poorly described the interaction of microbiota and gut barrier.
2. The abstract and main text cites the "omics" data but it is not well described in the main text. There are many manuscripts with single-cell data or even RNA-seq that have not been well discussed (e.g Integrated clinical and genomic analysis identifies driver events and molecular evolution of colitis-associated cancers - PubMed (nih.gov). In general, the review does not summarize the ffindings obtained using novel technologies or at least does not give the importance to these methods. It looks like a bit old-fashioned. I would suggest adding a figure with the impact of omics data in the information we can obtain from IBD and CRC.
3. There are some ideas overlapped through-out the text as first paragraph section 4 and Introduction.
4. The title is very broad. It should be modified. I think the abstract attracts the reader but then the main text does not really describe what the abstract says. I suggest the authors to focus on one aspect of the link between IBD and CRC and expand this as main objective.
Author Response
The manuscript by Nardone et al describes what is known about the influence of IBD in CRC progression. It also summarizes data from different therapies that are being used in IBD.
I found the manuscript quite interesting and well illustrated but:
- It is poorly described the interaction of microbiota and gut barrier.
Response: We thank the reviewer for the valuable suggestion. As suggested we have now included a paragraph and explained more in dept the interaction between microbiota-epithelial barrier-immune system. Page 7-8-9
- The abstract and main text cites the "omics" data but it is not well described in the main text. There are many manuscripts with single-cell data or even RNA-seq that have not been well discussed (e.g Integrated clinical and genomic analysis identifies driver events and molecular evolution of colitis-associated cancers - PubMed (nih.gov). In general, the review does not summarize the ffindings obtained using novel technologies or at least does not give the importance to these methods. It looks like a bit old-fashioned. I would suggest adding a figure with the impact of omics data in the information we can obtain from IBD and CRC.
Response: We have now included a paragraph on “OMICs and Future directions”. Page 4-5
- There are some ideas overlapped through-out the text as first paragraph section 4 and Introduction.
Response: Thank you for your comment. We have now cut down the overlapping concepts.
- The title is very broad. It should be modified. I think the abstract attracts the reader but then the main text does not really describe what the abstract says. I suggest the authors to focus on one aspect of the link between IBD and CRC and expand this as main objective.
Response: We have now changed the title as follows: “Inflammation-driven colorectal cancer associated with colitis - from pathogenesis to changing therapy”.
Reviewer 3 Report
The review written by Nardone and colleagues provides an overview how inflammatory bowel disease (IBD) and colorectal cancer are linked. The article is rather comprehensive by introducing the underlying molecular pathways of both pathologies, the implication of the microbiome and an extensive discussion on the cancer modulating effects of IBD targeting therapies. The review is suitable for publication in Cancers.
Minor points:
Primary sclerosing cholangitis should be introduced. Are additional pathologies known to be associated with IBD and to impact progression to colorectal cancer?
5-ASA compounds should be introduced.
Why does table 1 not include all current therapies, e.g. TNF antagonists? In this context: Is anything know on the effect of IL-6 targeting drugs on cancer?
The sentence in lines 253-255 is not clear.
Author Response
The review written by Nardone and colleagues provides an overview how inflammatory bowel disease (IBD) and colorectal cancer are linked. The article is rather comprehensive by introducing the underlying molecular pathways of both pathologies, the implication of the microbiome and an extensive discussion on the cancer modulating effects of IBD targeting therapies. The review is suitable for publication in Cancers.
Minor points:
Primary sclerosing cholangitis should be introduced. Are additional pathologies known to be associated with IBD and to impact progression to colorectal cancer?
Response: Thank you for your comment. We have now added an introduction: Primary sclerosing cholangitis (PSC) is a chronic cholestatic liver disease with a pooled prevalence of PSC in patients with IBD of 2.16%. Page 6
5-ASA compounds should be introduced.
Response: We have introduced 5-ASA as follows: Aminosalicylates (5-ASA), that include mesalazine (mesalamine), sulfasalazine, olsalazine and balsalazide are one of the oldest therapies currently used in the management of IBD. However, despite new treatment options have emerged, 5-ASA compounds remain effective first-line therapy for a step-up approach in treating mild to moderate UC and for maintenance of remission. Page 11
Why does table 1 not include all current therapies, e.g. TNF antagonists? In this context: Is anything know on the effect of IL-6 targeting drugs on cancer?
Response: As suggested, we edited the table adding the role of antiTNF drugs on CRC development. Additionally, IL6 pathway is mainly involved in autoimmune disease and it is considered a therapeutic target for these kind of diseases and cancers. For example, tocilizumab, an anti-IL6 receptor antibody, has shown to be effective in the treatment of autoimmune conditions, such as rheumatoid arthritis. However, to the best of our knowledge, there are no currently available drugs for IBD targeting IL6. Since our review is focused on current available therapies for IBD and their role on carcinogenesis, drugs targeting IL6 has not been considered. Page 15-16
The sentence in lines 253-255 is not clear.
Response: We have now rephrased it: “Probe confocal laser endomicroscopy (pCLE) is in vivo histology imaging tool capable of assessing ultrastructural and dynamic changes of the intestinal barrier consisting of epithelial cells connected by tight junctions and adherent junctions”
Round 2
Reviewer 2 Report
No more comments